# Herbicidal and Antibacterial Secondary Metabolites Isolated from the *Nicotiana tabacum*-Derived Endophytic Fungus *Aspergillus japonicus* TE-739D

**DOI:** 10.3390/plants14020173

**Published:** 2025-01-09

**Authors:** Haisu Wang, Xiaolong Yuan, Xinrong Huang, Peng Zhang, Gan Gu

**Affiliations:** Tobacco Research Institute of Chinese Academy of Agricultural Sciences, Qingdao 266101, China; haisu0815@163.com (H.W.); yuanxiaolong@caas.cn (X.Y.); hxr1352460520@163.com (X.H.)

**Keywords:** endophytic fungus, *Aspergillus japonicus*, secondary metabolites, polyketides, herbicidal activity, antibacterial activity

## Abstract

Endophytic fungi possess a unique ability to produce abundant secondary metabolites, which play an active role in the growth and development of host plants. In this study, chemical investigations on the endophytic fungus *Aspergillus japonicus* TE-739D derived from the cultivated tobacco (*Nicotiana tabacum* L.) afforded two new polyketide derivatives, namely japoniones A (**1**) and B (**2**), as well as four previously reported compounds **3**–**6**. Their chemical structures were elucidated by detailed spectroscopic analyses and quantum chemical calculations. In the herbicidal assays on the germination and radicle growth of *Amaranthus retroflexus* L. and *Eleusine indica* seeds, compound **1** was found to inhibit the germ and radicle elongation. Notably, compound **2** showed potent herbicidal activity against *A. retroflexus* L. germ elongation, with an IC_50_ value of 43.6 μg/mL, even higher than the positive control glyphosate (IC_50_ = 76.0 μg/mL). Moreover, compound **4** demonstrated strong antibacterial effects against the pathogens *Bacillus cereus* and *Bacillus subtilis*, with a comparable MIC value of 16 μg/mL to the positive control chloramphenicol. These findings indicate that the endophytic fungus *A. japonicus* TE-739D holds significant metabolic potential to produce bioactive secondary metabolites, which are beneficial, providing survival value to the host plants.

## 1. Introduction

Plant endophytic fungi are a group of microorganisms that live inside the plant tissues, bearing a symbiotic relationship with the host plants. Numerous studies have shown that plant endophytic fungi possess a unique ability to produce abundant secondary metabolites with attractive structures. In addition, these secondary metabolites are also reported to display a broad range of biological activities [1,2,3,4]. For instance, rhexocercosporin D, which was isolated from the endophytic fungus *Rhexocercosporidium* sp. Dzf14, could significantly reduce the growth of *Bacillus subtilis* and *Clavibacter michiganensis* with minimal inhibitory concentration (MIC) values of 4 and 2 µg/mL, respectively. Mechanistic studies showed that rhexocercosporin D could affect the homeostasis of a bacterial cell membrane to exert rapid bactericidal activity, which had potential as a lead compound for the creation of innovative antibacterials in crop disease management [5]. Monocillin I and radicicol, which were isolated from *Colletotrichum gloeosporioides* JS0419, showed potent in vitro antifungal activity against *Cryptococcus neoformans* H99, with an MIC value of 12.5 µM, equivalent to those of amphotericin B and fluconazole (12.5 µM), which are clinically used for the treatment of systemic cryptococcosis [6]. Sydonic acid isolated from the endophytic fungus *Aspergillus sparsus* NBERC_28952 showed obvious inhibitory effects on seedlings of *Amaranthus retroflexus* L., with an inhibitory rate of 78.34% at a concentration of 200 µg/mL, which can be used for developing natural herbicide products [7]. Moreover, these metabolites are proven to be beneficial to provide important survival value to the host plants, for example, by assisting hosts in the defense against pests and weeds [4]. The genus *Aspergillus* represents the most common and well-explored fungal species that distributes in various environments [8]. It is estimated that there are more than 378 species belonging to the genus *Aspergillus*, according to the World of Microorganisms Center [9]. Representative species of *Aspergillus* commonly found and studied include *A. flavus*, *A. fumigatus*, *A. nidurans*, *A. terreus*, *A. versicolor*, etc. [10]. The secondary metabolites isolated from these species have become the targets of medicines and agrochemicals due to their considerable biological activities [7,11].

Weeds are considered as undesirable plant species that occur in all types of agriculture, causing huge economic loss to crops, pastures, and rangelands [12,13]. The widespread use of chemical herbicides has played a positive role in the control of weeds. However, the long-term abuse of chemical herbicides, such asglyphosate, paraquat, prometryn, and trifluralin, has gradually caused resistance to weeds and environmental pollution problems [14], which emphasizes the importance of developing new naturally-sourced herbicides. During our ongoing search for natural herbicidal compounds from endophytic fungi [7,15,16], *Aspergillus japonicus* TE-739D, an endophyte isolated from the cultivated tobacco (*Nicotiana tabacum* L.), was chosen for further chemical investigations. As a result, two new polyketide derivatives **1** and **2,** as well as four known compounds **3**–**6,** were characterized (Figure 1). Compounds **1** and **2** were found to inhibit the germ and radicle elongation of weeds, suggesting high potential as candidates for weed management in agricultural settings. In this study, we present the structural elucidation of the novel compounds as well as their herbicidal and antibacterial activities.

## 2. Results

### 2.1. Structure Elucidation of Compounds

Compound **1** was obtained as a pale-yellow oil. Its molecular formula of C_14_H_22_O_5_ was identified by the high resolution electrospray ionization mass spectroscopy (HRESIMS) spectrum at *m*/*z* 271.1549 [M + H]^+^ (calcd. for C_14_H_23_O_5_, 271.1540), implying four degrees of unsaturation. The ^1^H NMR data (Table 1) of **1** showed four oxygenated methine protons at *δ*_H_ 4.33 (1H, ddt, *J* = 9.0, 7.5, 5.6 Hz, H-6), 4.22 (1H, t, *J* = 8.3 Hz, H-9), 4.01 (1H, br. s, H-12), and 3.75 (1H, d, *J* = 2.0 Hz, H-11), a terminal methyl group at *δ*_H_ 0.87 (3H, t, *J* = 6.8 Hz, H_3_-1), and several aliphatic protons (ranging from *δ*_H_ 1.28 to 2.40). In addition, three additional signals at *δ*_H_ 4.94 (br. s, OH-9), 4.80 (br. s, OH-11), and 5.09 (br. s, OH-12) were deduced to be exchangeable protons. The ^13^C NMR data (Table 1) combined with HSQC spectra revealed 14 carbon resonances, including an ester carbonyl at *δ*_C_ 165.1 (C-14), two olefinic carbons at *δ*_C_ 156.2 (C-8), and 123.1 (C-13), four oxygenated methine groups at *δ*_C_ 76.6 (C-6), 69.0 (C-11), 64.6 (C-12), and 64.4 (C-9), six methylene groups at *δ*_C_ 34.2 (C-5), 32.8 (C-10), 31.0 (C-3), 29.4 (C-7), 24.1 (C-4), and 22.1 (C-2), and one methyl at *δ*_C_ 13.9 (C-1).

The planar structure of compound **1** was determined by 2D NMR analyses (Figure 2). The ^1^H-^1^H COSY correlations of OH-9/H-9/H_2_-10/H-11/H-12/OH-12 and OH-11/H-11, as well as the HMBC correlations from H-9 to C-8/C-13, from H_2_-10 to C-8/C-9/C-11/C-12, from H-11 to C-12/C-13, and from H-12 to C-11 indicated the existence of a cyclohexene ring (ring A) with three hydroxyl groups attached to C-9/C-11/C-12. The HMBC correlations from H-6 to C-7/C-8/C-14 and from H_2_-7 to C-6/C-8/C-13 confirmed that the lactonic ring (ring B) was fused with ring A at C-8/C-13. Finally, the establishment of the alkane side chain C-1–C-5 attached to C-6 was revealed by ^1^H-^1^H COSY correlations of H_3_-1/H_2_-2/H_2_-3/H_2_-4/H_2_-5/H-6 and the key HMBC correlations from H_2_-4 to C-6, from H-6 to C-4/C-5, and from H_2_-7 to C-5. Thus, the planar structure of **1** was constructed as shown in Figure 1.

Since ring A represents a typical cyclohexane structure, the relative configurations of **1** were determined by NOESY spectrum (Figure 2) and *J* coupling constants. A small coupling constant of H-11 (*J* = 2.0 Hz), as well as the NOE correlations between H-11/H-10eq (*δ*_H_ 1.90, m) and H-11/H-10ax (*δ*_H_ 1.82, ddd, *J* = 12.6, 10.1, 2.0 Hz), suggested H-11 was an equatorial bond (tentatively assigned as *β* orientation). Moreover, the large coupling constant of H-9 (*J* = 8.3 Hz) suggested that H-9 was an axial bond. H-12 was identified as *β* orientation by the NOE correlations of H-12/H-11 and H-12/OH-9. However, the relative configuration of C-6 could not be determined based on the above NMR data. In order to address the configuration of C-6, the GIAO ^13^C NMR calculations at the mpw1pw91/6-311 + g(2d,p) (PCM = DMSO)//B3LYP/6-31 + G(d,p) level of theory using the reported procedure and scaling parameters [17] revealed that 6*S**-**1** fit well to the experimental data, possessing smaller MAE/RMSD/R^2^ values (0.8/1.8/0.9991) than those of 6*R**-**1** (1.1/2.1/0.9981) (Table 2, Figure 3). Finally, the DFT-calculated ECD spectra of 6*S*,9*S*,11*S*,12*R*-**1** and the enantiomer of 6*R*,9*R*,11*R*,12*S*-**1** were obtained at the B3LYP/6-31 + G(d) level. The experimental ECD curve was in accordance with the calculated ECD spectrum of **1**, which ascertained the absolute configurations of **1** (Figure 4).

Compound **2** had the same molecular formula of C_14_H_22_O_5_ as compound **1,** according to its HRESIMS. The analysis of the ^13^C NMR data (Table 1) showed that the structure of **2** was closely related to that of **1**. The main difference was the absence of an ester carbonyl group rather than an additional ketone group in **2**. The key HMBC correlations (Figure 5) from H-12 to C-8/C-10/C-11/C-13, from H_2_-10 to C-8/C-9/C-12, and from H-11 to C-9/C-10/C-12, as well as the ^1^H-^1^H COSY correlations of H_2_-10/H-11/H-12 established a 4,5-dihydroxy-2-cyclohexen-1-one ring system (ring A). Ring B was identified by the ^1^H-^1^H COSY correlations of H-6/H_2_-7 and HMBC correlations from H_2_-7 to C-8/C-13 and from H-14 to C-6/C-8/C-13 (Figure 5). Similarly, the alkane side chain C-1–C-5 attaching to C-6 was supported by the ^1^H-^1^H COSY correlations of H_3_-1/H_2_-2/H_2_-3/H_2_-4/H_2_-5/H-6 and the HMBC correlations from H_2_-5 to C-6/C-7 and from H_2_-7 to C-5/C-6. However, due to the lack of key HMBC correlations, two candidate structures (**2a** and **2b**) existed (Figure 5). The coupling constants of H-6 (*J*_6,7_ = 11.1 Hz) and H-11 (*J*_11,12_ = 6.1 Hz) suggested that the H-6, H-11, and H-12 were axially orientated. There was no NOE correlation between H-6 and H-14, indicating that H-14 was equatorial. The key NOE correlation of H-12/H-14 (Figure 6) indicated that both groups were close in three-dimensional space, suggesting that the structure of **2a** was more reasonable. This deduction was verified by the GIAO ^13^C NMR calculations and the experimentally observed NMR data for **2**, which gave the best match to that of the **2a1** with smaller MAE/RMSD/R^2^ values (0.8/1.4/0.999) than **2b1** (1.6/3.2/0.995) (Appendix A). The absolute configurations of **2** were determined to be 6*S*,11*R*,12*R*,14*R* by the calculated ECD spectrum at the B3LYP/6-31 + G(d)//B3LYP/6-311 + G(d,p) level of theory (Figure 7).

In addition, the known compounds were identified based on the literature data as 3*β*,15*β*-dihydroxy-(22*E*,24*R*)-ergosta-5,8(14),22-trien-7-one (**3**) [18], 3*β*,15*α*-dihydroxyl-(22*E*,24*R*)-ergosta-5,8(14),22-trien-7-one (**4**) [18], honokiol (**5**) [19], and eupenoxide (**6**) [20].

### 2.2. Herbicidal Activities of Compounds ***1*** and ***2***

Compounds **1** and **2** (at the concentration of 200 μg/mL) were tested for their effects on the germination and radicle growth of *A. retroflexus* L. and *Eleusine indica* seeds. Compound **1** exhibited a substantial inhibitory effect on the radicle growth of *E. indica*, achieving an inhibition rate of 85.6%. Moreover, compound **1** demonstrated good inhibitory effects on the germ and radicle elongation of *A. retroflexus* L., with inhibition rates of 73.0% and 80.7%, respectively (Figure 8A). Compound **2** also showed remarkable inhibitory effects on the germination and radicle growth of *A. retroflexus* L., with respective inhibition rates of 85.6% and 85.9%. Interestingly, compound **2** had a weak effect on *E. indica* germ elongation with an inhibition rate of 16.8%, while it could significantly promote the elongation of the radicle. Due to the limitation of the compound amounts, the IC_50_ values of compound **1** for *E. India* and compound **2** for *A. retroflexus* L. germ and radicle elongation effects were measured. The IC_50_ values of compound **1** for inhibiting the radicle elongation of *E. indica* seeds was 97.7 μg/mL (Figure 8B). Compound **2** showed strong herbicidal activity against *A. retroflexus* L. germ elongation, with an IC_50_ value of 43.6 μg/mL, which was more effective than the positive control glyphosate (IC_50_ = 76.0 μg/mL) (Figure 8C).

### 2.3. Antibacterial Activities of Compounds ***1**–**6***

The antibacterial activities of compounds **1**–**6** were evaluated against four bacterial strains: *Ralstonia solanacearum* (G^−^), *Xanthomonas oryzae* (G^−^), *Bacillus cereus* (G^+^), and *Bacillus subtilis* (G^+^). As presented in Table 3, compound **5** demonstrated strong inhibitory effects against *B. cereus* and *B. subtilis*, with a minimum inhibitory concentration (MIC) value of 16 μg/mL for both strains, comparable to the positive control chloramphenicol. Additionally, compound **4** exhibited moderate inhibitory effects against *B. cereus* and *B. subtilis*, with MIC values of 32 μg/mL and 16 μg/mL, respectively. Notably, compound **4** differed from compound **3** only in chirality at C-15, yet it exhibited significantly higher antibacterial activity, suggesting its potential as a chiral drug in pharmaceuticals [21,22].

## 3. Discussion

Weeds are considered to be the major factor in crop yield reduction. Following the progress of social civilization and the improvement in public health awareness, the negative effects of the widespread, long-term, and large-scale use of chemical herbicides are becoming increasingly apparent. This urgently requires us to explore new herbicides with novel mechanisms of action [12,14]. Natural products and organisms engage in co-evolution within ecosystems, influencing the metabolic pathways of specific cells and thereby playing a crucial role in agricultural production. For instance, cinnamic acid methyl ester has been identified as a selective suppressor of annual ryegrass (*Lolium rigidum*) in wheat cultivation [23]. Additionally, the fungal phytotoxin radicinin exhibits selectivity against buffel grass (*Cenchrus ciliaris* L.), an invasive species in the United States, while demonstrating no teratogenic, toxic, or lethal effects on Brachydanio rerio embryos [24]. Furthermore, the potent toxin diacetoxyscirpenol has been shown to completely inhibit the germination of *Striga hermontica* (Delile) Benth. and *Orobanche ramosa* L. seeds at concentration below 1 µM [25,26]. Therefore, natural products have great potential for application in herbicide development.

In this study, two novel compounds with strong herbicidal activity were isolated from the endophytic fungus *A. japonicus* TE-739D. Compound **1** demonstrated significant inhibitory effects on both *A. retroflexus* L. and *E. indica*, suggesting its potential as a natural herbicide. This broad-spectrum inhibitory effect on seed germination may be attributed to its ability to disrupt essential physiological processes in weed seeds, such as inhibiting enzymes involved in germination or interfering with hormonal regulation. It should be pointed out that the concentrations of the tested compounds (200 μg/mL for seed germination inhibition assays) are not biologically relevant. The yield of these metabolites produced in situ is very low, far from the concentration required for the experiments. It is unable to detect the true concentrations of these compounds within the endophytes. Herein, we merely tested the herbicidal activities of these compounds in accordance with the bioactivity testing methods, with the aim of identifying potential lead molecules with agricultural applications. Moreover, compound **2** exhibited strong inhibitory activity against *A. retroflexus* L., while paradoxically promoting radicle elongation in *E. indica*. This differential response may be attributed to variations in the physiological structures and metabolic pathways of the two weed species, which likely influence their absorption, transformation, and response mechanisms to the compound, resulting in inhibition in one species and promotion in the other. The genomic data of *A. retroflexus* L. have recently been available [27], providing a valuable resource for accurately identifying the targets of various compounds. It is necessary to choose some model plants, such as Arabidopsis, tomato, rice, and wheat, to eliminate the toxic effects on cultivated plants. This advancement will facilitate a deeper understanding of how these compounds influence weed physiology, enable the exploration of their metabolic pathways within weeds, and allow for the assessment of their efficacy and persistence. Such insights are crucial for ensuring ecological security and supporting the rational use of herbicides. However, due to the current limitations in compound availability, the present study was confined to assays assessing seed germination inhibition. Most importantly, this study focused on the herbicidal activity of the fungal polyketides against two weeds, ignoring the phytotoxic effects on the host plant *N. tabacum*. It would be interesting to discuss the potential phytotoxicity of these metabolites on non-target plant species. Future research will focus on the chemical synthesis of these compounds to further investigate their herbicidal activities in both controlled and field environments, as well as to evaluate their potential phytotoxic effects on non-target plant species.

An interesting result appeared in the antibacterial activity test. Compound **4** showed good antibacterial activity against two Gram-positive bacteria, unlike compound **3**, which is inactive. The only difference between them is chirality at C-15, highlighting how crucial chirality is for antibacterial properties. This insight can guide pesticide development by focusing on chiral centers to enhance the effectiveness and reduce ineffective isomers, thereby lowering the costs and environmental risks. Understanding the role of chirality in antibacterial mechanisms can also optimize pesticide design and deepen our understanding of antibacterial principles. In addition, we chose our existing bacterial strains, the two closely related species of *B. cereus* and *B. subtilis,* for antibacterial assays in this study. Since the cell wall composition is often crucial for antimicrobial effects, further work should test more bacteria species to fully evaluate the antimicrobial effects of these compounds.

## 4. Materials and Methods

### 4.1. General Experimental Procedures

UV spectra were obtained using a Techcomp UV2310II spectrophotometer (Techcomp, Ltd., Shanghai, China). Specific rotations were recorded on a Rudolph Autopol IV automatic polarimeter (Rudolph Research Analytical, Flanders, NJ, USA). Circular dichroism (CD) spectra were recorded on a JASCO J-1500 CD spectrometer (JASCO Corp., Tokyo, Japan). High-resolution electrospray ionization mass spectrometry (HRESIMS) spectra were recorded on a Micromass Q-TOF spectrometer (Waters, Milford, MA, USA). ^1^H, ^13^C, and 2D NMR (HSQC, HMBC, ^1^H-^1^H COSY, NOESY) spectra were measured on a Bruker Avance NEO 500 NMR spectrometer (Bruker BioSpin, Zürich, Switzerland) and an Agilent DD2 500 MHz NMR spectrometer (Agilent Technologies, Santa Clara, CA, USA). Chemical shifts are expressed in *δ* (ppm) referring to the solvent residual peaks at *δ*_H_ 2.50 and *δ*_C_ 39.5 for DMSO-*d*_6_ and *δ*_H_ 3.31 and *δ*_C_ 49.0 for CD_3_OD, and the coupling constants (*J*) are in hertz (Hz). Column chromatography (CC) was performed on silica gel (200−300 mesh; Qing Dao Hai Yang Chemical Group Co., Qingdao, China), Sephadex LH-20 (GE Healthcare, Pittsburgh, PA, USA), and reversed-phase (RP)-18 gel (40–60 μm, Merck, Darmstadt, Germany). Semipreparative HPLC separation was carried out on an NS4201 instrument (Hanbon Sci.& Tech., Huaiyin, China) equipped with an NP7000 SERIALS pump (flow rate: 3 mL/min) and an NU3000 SERIALS UV detector using a SunFire-C_18_ column (150 mm × 10 mm i.d., 5 μm, Waters, Milford, MA, USA).

### 4.2. Separation, Purification, and Identification of A. japonicus TE-739D

The separation and purification of *A. japonicus* TE-739D was performed based on previously reported methods with certain modifications [28]. The clean leaves of *N. tabacum* (grown in Enshi, Hubei Province, People’s Republic of China) were sterilized with 75% ethanol for 2 min, immersed successively in 2.5% sodium hypochlorite for 20 min, and then rinsed in sterile distilled water three times. The potato dextrose agar (PDA) plate was used to provide nutrients for leaves explants, which were cut into small pieces of 0.5 cm × 0.5 cm with a scalpel, and 2–3 pieces were placed on each PDA plate containing 500 µg/mL of streptomycin sulfate. Finally, the inoculated plates were incubated at 28 °C for 7–14 days in the dark. The pure cultures were isolated by hyphal tip isolation on PDA plates until the colony morphology was stable and consistent.

Based on the phylogenetic analyses of the 26S and internal transcribed spacer (ITS) rDNA regions, this fungus was identified as *Aspergillus japonicus*, which has been deposited in GenBank (NCBI) with the number of PP126510. The strain was deposited in the China General Microbiological Culture Collection Center (CGMCC No. 40901).

### 4.3. Fermentation and Extraction

The fungus was statically cultivated in 290 × 1 L Erlenmeyer flasks at 28 °C for 30 days in Potato Dextrose Water medium. After cultivation, the fermentation product was extracted with ethyl acetate (EtOAc), and the crude extract (68.8 g) was obtained with evaporation under vacuum.

### 4.4. Isolation

The extract was eluted with stepwise petroleum ether (PE)−EtOAc mixtures (100:0, 90:10, 80:20, 70:30, 50:50, 30:70, and 0:100, *v*/*v*) to yield Fr.1−Fr.6. Fr.6 (eluted with PE−EtOAc 0:100) (13.8 g) was further separated using reverse silica gel column chromatography (CC) with a gradient elution of MeOH−H_2_O (30:70−100:0, *v*/*v*) to yield seven subfractions (Fr.6.1−Fr.6.7). Fr.6.3 was then separated by semipreparative HPLC using MeCN−H_2_O (17:83, *v*/*v*) as an eluent to afford compound **1** (4.3 mg, *t*_R_ 19.868 min). Similarly, compounds **3** (22.5 mg, *t*_R_ 28.893 min) and **4** (5.6 mg, *t*_R_ 30.943 min) were obtained from Fr.5 by semipreparative HPLC with MeCN−H_2_O (77:23, *v*/*v*). Compound **6** (3.2 mg, *t*_R_ 43.917 min) was yielded by semipreparative HPLC with MeCN−H_2_O (40:60, *v*/*v*) from Fr.4. Compounds **2** (3.1 mg, *t*_R_ 12.674 min) and **5** (8.3 mg, *t*_R_ 18.019 min) were isolated from Fr.6.4 using Sephadex LH-20 CC (MeOH) and semipreparative HPLC eluted with 32% MeOH in H_2_O.

### 4.5. Spectral Data of the Isolated Compounds

#### 4.5.1. Japonione A (**1**)

Pale-yellow oil; [α]^24^_D_ +10.7 (*c* 0.27, MeOH); ECD (*c* = 1.1 × 10^−3^ M, MeOH) *λ* 213, 238, 265 nm; UV (MeOH) *λ*_max_ 211.1 nm; ^1^H NMR (DMSO-*d*_6_, 500 MHz), ^13^C NMR (DMSO-*d*_6_, 125 MHz), see Table 1; HRESIMS *m*/*z* 271.1549 [M + H]^+^ (calcd. for C_14_H_23_O_5_, 271.1540).

#### 4.5.2. Japonione B (**2**)

Yellow oil; [α]^24^_D_ +55.0 (*c* 0.08, MeOH); ECD (*c* = 1.2 × 10^−3^ M, MeOH) *λ* 240, 296, 344 nm; UV (MeOH) *λ*_max_ 235.9 nm; ^1^H NMR (CD_3_OD, 500 MHz), ^13^C NMR (CD_3_OD, 125 MHz), see Table 1; HRESIMS *m*/*z* 269.1398 [M − H]^−^ (calcd. for C_14_H_21_O_5_, 269.1394).

### 4.6. Seed Germination Inhibition Assays

Two weed seeds (*Amaranthus retroflexus* L. and *Eleusine indica*) were used to evaluate the herbicidal activity of compounds **1** and **2**, based on previously reported methods with certain modifications [7,29]. The seeds were soaked in water for 8 h, then sterilized with 4% sodium hypochlorite for 10 min, and repeatedly rinsed in sterile water. Meanwhile, the tested compounds were dissolved in 200 μL methanol (200 μg/mL) in 12-well plates containing sterile filter paper at the bottom of each well. After evaporation of the solvents, ten weed seeds were put into each hole with 200 μL sterile water. Then, the 12-well plate was sealed with parafilm and incubated in the dark at 25 °C for 96 h. Each treatment was run twice, and glyphosate and sterile water (200 μL per well) were used as the positive and blank controls, respectively. Finally, the germ and radicle lengths of each seed were measured, and the seed germination inhibition rates were calculated using the following equation:inhibition rate (%) = (*L*_c_ − *L*_t_)/*L*_c_ × 100,
where *L*_c_ and *L*_t_ are the lengths (radicle and germ) of the control and treatment groups, respectively.

### 4.7. Antibacterial Assays

The antibacterial activities of compounds **1**–**6** were evaluated against two Gram-positive bacteria (*Bacillus cereus* and *Bacillus subtilis*) and two Gram-negative bacteria (*Ralstonia solanacearum* and *Xanthomonas oryzae*). Chloramphenicol was used as a positive control. The minimum inhibitory concentrations (MICs) of each compound were determined by the broth micro-dilution method with certain modifications [5,30]. The compounds were initially dissolved in DMSO in 5.12 mg/mL, and then 2.5 μL of compound solution mixed with 97.5 μL of bacterial suspensions was added in Luria-Bertani (LB) medium containing approximately 10^6^ CFU/mL in a sterilized 96-well microtiter plate. Finally, the MICs were determined as the concentrations of compounds where no bacterial growth was visible after incubation for 18 h at 37 °C for *B. cereus* and *B. subtilis* and 28 °C for *R. solanacearum* and *X. oryzae*.

### 4.8. Statistical Analysis

Each treatment was conducted in triplicate, and the inhibition rates of the tested weeds were analyzed using IBM SPSS Statistics 27 with one-way analysis of variance. All experimental data were expressed as the mean ± standard deviation, with statistical significance set at *p* < 0.05.

## 5. Conclusions

In conclusion, this study reported chemical studies on an *N. tabacum*-derived endophytic fungus *A. japonicus* TE-739D, which led to the isolation and identification of two new polyketide derivatives **1** and **2**. The structures of the new compounds were determined by HRESIMS, NMR spectroscopic data, and calculated ECD spectra. The herbicidal effects of two new compounds were tested on *E. indica* and *A. retroflexus* L. Japonione A (**1**) effectively inhibited the germination and radicle growth of both weeds. Japonione B (**2**) significantly reduced the radicle length in *A. retroflexus* L, and its IC_50_ was higher than glyphosate. Moreover, the known compound **4** displayed strong antibacterial effects against the pathogens *B. cereus* and *B. subtilis*. These findings provide a solid foundation for the metabolic potential of the endophytic fungus, which are beneficial for understanding the symbiotic relationships between endophytes and their host plants.

## Figures and Tables

**Figure 1 plants-14-00173-f001:**
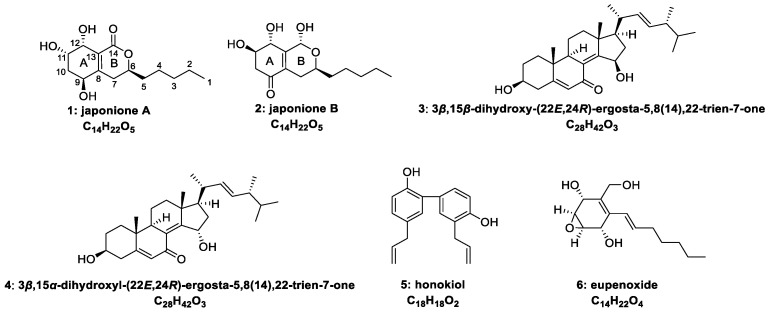
Structures of the isolated compounds **1**–**6**.

**Figure 2 plants-14-00173-f002:**
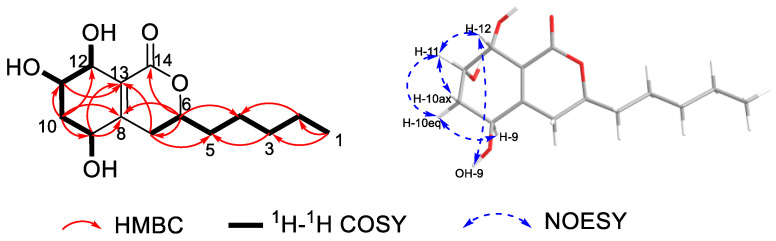
Key ^1^H–^1^H COSY, HMBC, and NOESY correlations of compound **1**.

**Figure 3 plants-14-00173-f003:**
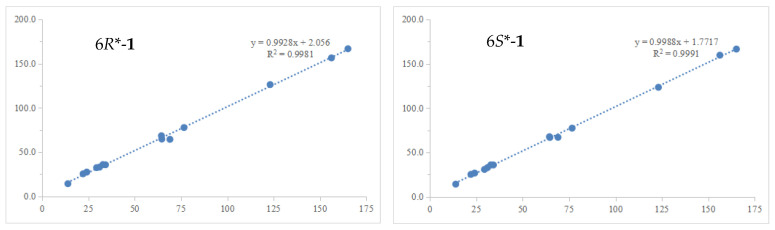
Linear regression between Exp. *δ*_C_ (x) and Calc. *δ*_C_ (y) of 6*R**-**1** and 6*S**-**1**.

**Figure 4 plants-14-00173-f004:**
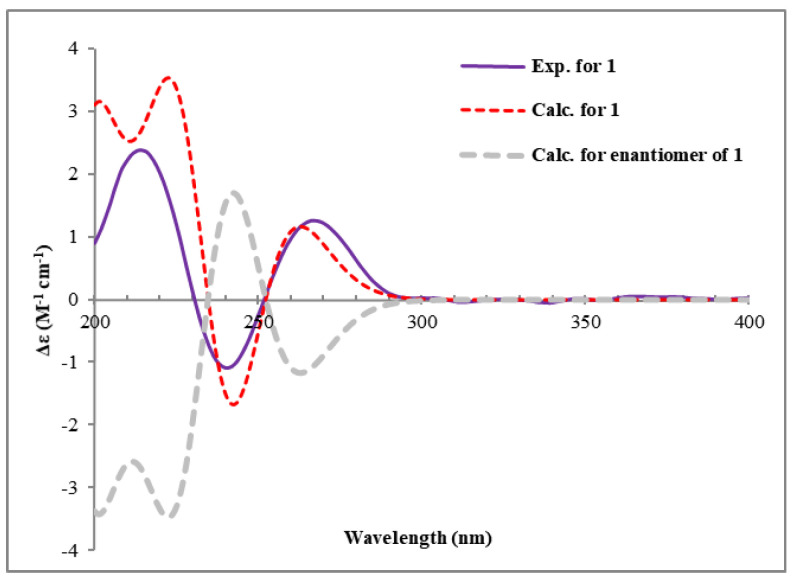
Calculated and experimental ECD spectra of compound **1**.

**Figure 5 plants-14-00173-f005:**
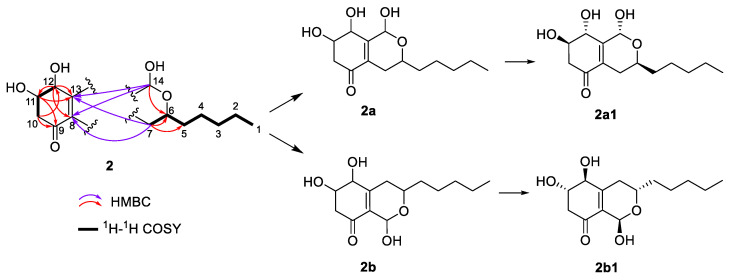
Selected HMBC and ^1^H–^1^H COSY correlations of **2**, its two possible planar structures (**2a** and **2b**), and the related stereomers that fit the NOESY correlations and were used for ^13^C NMR calculations.

**Figure 6 plants-14-00173-f006:**
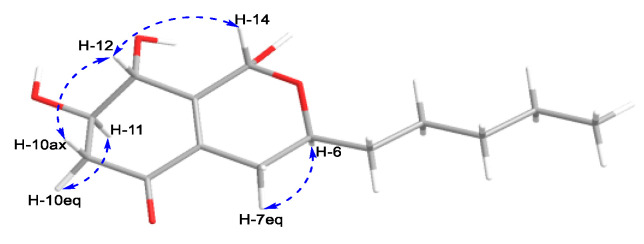
Key NOE correlations (blue dashed arrows) of compound **2**.

**Figure 7 plants-14-00173-f007:**
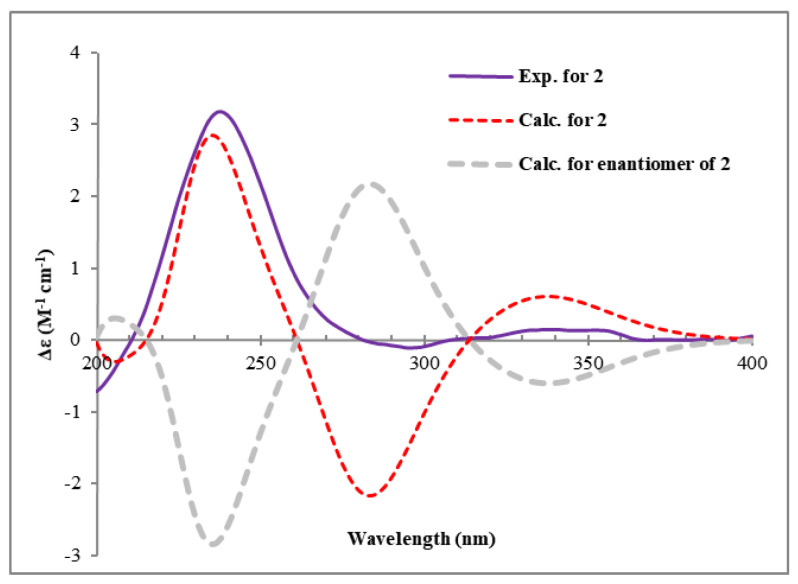
Calculated and experimental ECD spectra of compound **2**.

**Figure 8 plants-14-00173-f008:**
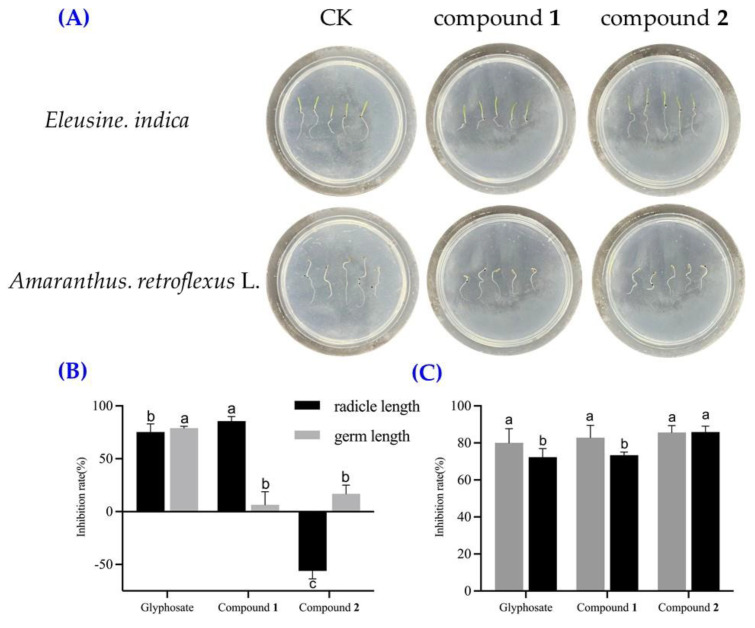
(**A**) *Eleusine indica* and *Amaranthus retroflexus* L. seedlings were treated with 200 μg/mL of compounds for 4 days; CK, blank control with sterile water. (**B**) Inhibition effects of compounds **1** and **2** on seed germination of *E. Indica*. (**C**) Inhibition effects of compounds **1** and **2** on seed germination of *A. retroflexus* L. Different letters in (**B**) or (**C**) indicate significant differences according to Duncan’s multiple range test (*p ≤* 0.05).

**Table 1 plants-14-00173-t001:** ^1^H NMR data (500 MHz) and ^13^C NMR data (125 MHz) for compounds **1** and **2**.

Position	1 ^a^		2 ^b^	
	*δ*_H_ Mult. (*J* in Hz)	*δ*_C_ Type	*δ*_H_ Mult. (*J* in Hz)	*δ*_C_ Type
1	0.87 t (6.8)	13.9 CH_3_	0.91 t (6.9)	14.4 CH_3_
2	1.29 m	22.1 CH_2_	1.36 m	23.7 CH_2_
3	1.28 m	31.0 CH_2_	1.34 m	32.9 CH_2_
4	1.38 m 1.32 m	24.1 CH_2_	1.51 m 1.43 m	26.1 CH_2_
5	1.67 ddt (13.7, 8.3, 5.2) 1.58 ddt (13.7, 10.9, 5.6)	34.2 CH_2_	1.55 m	36.4 CH_2_
6	4.33 ddt (9.0, 7.5, 5.6)	76.6 CH	3.89 ddt (11.1, 7.9, 4.1)	67.3 CH
7	2.40 m	29.4 CH_2_	2.34 ddd (17.6, 3.8, 2.2) 1.79 dd (17.6, 11.1)	28.4 CH_2_
8		156.2 C		133.0 C
9	4.22 t (8.3)	64.4 CH		198.9 C
10	1.90 m 1.80 ddd (12.6, 10.1, 2.0)	32.8 CH_2_	2.83 dd (16.3, 4.2) 2.45 dd (16.3, 8.4)	43.8 CH_2_
11	3.75 d (2.0)	69.0 CH	3.98 ddd (8.4, 6.1, 4.2)	72.4 CH
12	4.01 br. s	64.6 CH	4.30 dt (6.1, 2.2)	71.0 CH
13		123.1 C		152.3 C
14		165.1 C	5.67 s	89.2 CH
9-OH	4.94 br. s			
11-OH	4.80 br. s			
12-OH	5.09 br. s			

^a^ In DMSO-*d*_6_. ^b^ In CD_3_OD.

**Table 2 plants-14-00173-t002:** GIAO ^13^C NMR calculation of **6*R**-1** and **6*S**-1**.

Position	Exp. *δ*_C_	Cal. *δ*_C_ ^a^ (6*R**-1)	Δ*δ* (Cal. − Exp.)	Cal. *δ*_C_ ^a^ (6*S**-1)	Δ*δ* (Cal. − Exp.)
1	13.9	14.2	0.3	14.2	0.3
2	22.1	25.4	3.3	25.3	3.2
3	31	33.0	2.0	32.8	1.8
4	24.1	27.4	3.3	26.7	2.6
5	34.2	35.6	1.4	35.9	1.7
6	76.6	77.7	1.1	77.3	0.7
7	29.4	32.4	3.0	30.8	1.4
8	156.2	156.5	0.3	159.7	3.5
9	64.4	68.5	4.1	67.7	3.3
10	32.8	35.9	3.1	35.9	3.1
11	69.0	64.4	−4.6	67.0	-2.0
12	64.6	64.7	0.1	66.9	2.3
13	123.1	126.3	3.2	123.4	0.3
14	165.1	166.8	1.7	166.6	1.5
		RMSD	2.1	RMSD	1.8
		MAE	1.1	MAE	0.8

^a 13^C NMR calculations were performed at the mpw1pw91/6-311 + G(2d,p) (PCM = MDSO)//B3LYP/6-31 + G(d,p) level; then, cal. *δ*_C_ was scaling corrected from the calculated shielding tensor (*σ*) using the formula *δ*_C_ = (186.2534-*σ*)/1.0496.

**Table 3 plants-14-00173-t003:** Antibacterial activities of the isolated compounds **1**–**6** (MIC, μg/mL).

Compound	*Ralstonia* *solanacearum*	*Xanthomonas oryzae*	*Bacillus* *cereus*	*Bacillus subtilis*
**1**	>128	>128	>128	>128
**2**	>128	>128	>128	>128
**3**	>128	>128	>128	>128
**4**	>128	>128	32	16
**5**	>128	>128	16	16
**6**	>128	>128	>128	>128
Chloramphenicol ^a^	16	4	16	16

^a^ Positive control.

## Data Availability

Data are contained within the article and Appendix A.

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
