# Peer review of "Herbicidal and Antibacterial Secondary Metabolites Isolated from the Nicotiana tabacum-Derived Endophytic Fungus Aspergillus japonicus TE-739D"

_plants, 2025, doi:10.3390/plants14020173_

Round 1
Reviewer 1 Report
Comments and Suggestions for Authors
Are the concentrations of the tested endophyte compounds biologically relevant, i.e. are they produced in situ to reach these concentrations?
The effects on two plant species by the fungal polyketide were tested but not on the host plant Nicotiana tabacum. Why?
The description of the results on the inhibition of radicle length and germ elongation is highly misrepresenting. Compound 2 is highly STIMULATORY on radicle length in E. indica. Why exclude this result; maybe this is the biologically relevant effect by a fungal symbiont, rather than the opposite inhibition, as the authors seem to imply? Tables S2 and S3 (lacking experiments) are also incomplete in a similar way.
The antimicrobial effects on Bacillus is potentially interesting but maybe not relevant for grampositive bacteria as such. Why test two closely related species as representatives for G+-bacteria? The peptidoglycan cell wall of Bacillus is aberrant compared to the majority of the grampositives; why not at least use one species with lys-linked and one with DAP-linked peptidoglycans in their walls. Cell wall composition is often crucial for antimicrobial effects.
Author Response
Comment 1: Are the concentrations of the tested endophyte compounds biologically relevant, i.e. are they produced in situ to reach these concentrations?
Response: Thank you for your helpful comments. The concentrations of the tested compounds are not biologically relevant. These compounds could not be produced in situ to achieve the tested concentrations (i.e. 200 μg/mL for seed germination inhibition assays). Indeed, although endophytes can produce abundant secondary metabolites, the yield of these metabolites is very low, far from the concentration required for the experiments. We are unable to detect the true concentrations of these compounds within the endophytes. Herein, we merely tested the herbicidal and antibacterial activities of these compounds in accordance with the bioactivity testing methods, with the aim of identifying potential lead molecules with agricultural applications. We have added the above discussions in the Discussion part (Lines 224-231).
“It should be pointed out that the concentrations of the tested compounds (200 μg/mL for seed germination inhibition assays) are not biologically relevant. The yield of these metabolites produced in situ is very low, far from the concentration required for the experiments. It is unable to detect the true concentrations of these compounds within the endophytes. Herein, we merely tested the herbicidal activities of these compounds in accordance with the bioactivity testing methods, with the aim of identifying potential lead molecules with agricultural applications.”
Comment 2: The effects on two plant species by the fungal polyketide were tested but not on the host plant Nicotiana tabacum. Why?
Response: Thank you for your helpful comments. Yes, our study focused on the herbicidal activity of the fungal polyketides against two weeds. However, the effects on the host plant Nicotiana tabacum have been neglected. It is interesting to discuss the potential phytotoxicity of these metabolites on non-target plant species. We appreciate the reviewer for reminding us this important issue. We have added the following discussions in the Discussion part (Lines 245-251). “Most importantly, this study focused on the herbicidal activity of the fungal polyketides against two weeds, ignoring the phytotoxic effects on the host plant N. tabacum. It is interesting to discuss the potential phytotoxicity of these metabolites on non-target plant species. Future research will focus on the chemical synthesis of these compounds to further investigate their herbicidal activities in both controlled and field environments, as well as to evaluate their potential phytotoxic effects on non-target plant species.”
Comment 3: The description of the results on the inhibition of radicle length and germ elongation is highly misrepresenting. Compound 2 is highly STIMULATORY on radicle length in E. indica. Why exclude this result; maybe this is the biologically relevant effect by a fungal symbiont, rather than the opposite inhibition, as the authors seem to imply? Tables S2 and S3 (lacking experiments) are also incomplete in a similar way.
Response: Thank you for your helpful comments. Yes, we totally agree that compound 2 has a promoting effect on E. indica radicle length, which may be caused by the biologically relevant effect by a fungal symbiont. As suggested, we have added this results in the manuscript (Lines 176-177. Interestingly, compound 2 had a weak effect on E. indica germ elongation with inhibition rate of 16.8%, while it could significantly promote the elongation of the radicle.). Moreover, we have added the possible reasons for such differences in the discussion part (Lines 231-236. Moreover, compound 2 exhibited strong inhibitory activity against A. retroflexus L., while paradoxically promoting radicle elongation in E. indica. This differential response may be attributed to variations in the physiological structures and metabolic pathways of the two weed species, which likely influence their absorption, transformation, and response mechanisms to the compound, resulting in inhibition in one species and promotion in the other.). As for the lacking experiments in Tables S2 and S3, due to the limitation of compounds amount, we only tested the IC50 values of compound 1 for E. India and compound 2 for A. retroflexus L. germ and radicle elongation effects. We have added this information in revised manuscript (Lines 178-180).
Comment 4: The antimicrobial effects on Bacillus is potentially interesting but maybe not relevant for gram-positive bacteria as such. Why test two closely related species as representatives for G+-bacteria? The peptidoglycan cell wall of Bacillus is aberrant compared to the majority of the gram-positives; why not at least use one species with lys-linked and one with DAP-linked peptidoglycans in their walls. Cell wall composition is often crucial for antimicrobial effects.
Response: Thank you for your helpful comments. The antibacterial assays have been performed based on our existing bacterial strains, Ralstonia solanacearum (G-), Xanthomonas oryzae (G-), Bacillus cereus (G+), and Bacillus subtilis (G+). We did not consider the impact of peptidoglycan cell wall of Bacillus on antibacterial activity. We appreciate the reviewer for reminding us this important issue. We have added some discussions on this issue in the Discussion part (Lines 259-262). “In addition, we chose our existing bacterial strains, the two closely related species of B. cereus and B. subtilis for antibacterial assays in this study. Since the cell wall composition is often crucial for antimicrobial effects, further work should add more bacteria species to fully evaluate the antimicrobial effects of these compounds.”
Reviewer 2 Report
Comments and Suggestions for Authors
Herbicidal and Antibacterial Secondary Metabolites Isolated from the Nicotiana tabacum-Derived Endophytic Fungus Aspergillus japonicus TE-739D
By Haisu Wang , Xiaolong Yuan , Xinrong Huang , Peng Zhang * , Gan Gu *
This study is devoted to a detailed description of biologically active compounds isolated from the tobacco endophytic fungus Aspergillus japonicus TE-739D and an analysis of their herbicidal and antibacterial activity. The authors studied the chemical structure of the compounds in detail. However, I have several suggestions regarding the herbicidal and antibacterial activity.
The main topic of criticism:
1. To analyse the herbicidal activity, the authors chose two weeds. As a control, it is necessary to choose any model plant - Arabidopsis, tomato, rice, wheat, etc. This is necessary to eliminate the toxic effect on cultivated plants. Only then can we talk about the herbicidal activity of the substances under investigation.
2. Glyphosate was also used as a control when analysing herbicidal activity (Fig.8b,c). It is also necessary to show how plants grew on media with no substances added - a control is needed. There is a control in Figure 8, but what is it? Regular medium or medium with added glyphosphate?
3. Please provide figures on the inhibitory activity of the tested substances against bacteria.
Does the endophytic fungus Aspergillus japonicus TE-739D have herbicidal and antibacterial properties? Why don't you use an endophytic fungus as an additional control?
- Minor Revisions
see the pdf file
Decision: - major revisions.

Author Response
Comment 1: To analyse the herbicidal activity, the authors chose two weeds. As a control, it is necessary to choose any model plant - Arabidopsis, tomato, rice, wheat, etc. This is necessary to eliminate the toxic effect on cultivated plants. Only then can we talk about the herbicidal activity of the substances under investigation.
Response: Thank you for your helpful comments. Yes, we totally agree that it is necessary to choose some model plants to eliminate the toxic effect on cultivated plants. We appreciate the reviewer for reminding us this important issue. However, due to the limitation of compounds amount, it is impossible for us to supplement this experiment. Therefore, we have added some discussions on this issue in the Discussion part, which hope to get your approval (Lines 236-243).
“The genomic data of A. retroflexus L. has recently been available, providing a valuable resource for accurately identifying the targets of various compounds. It is necessary to choose some model plants, such as Arabidopsis, tomato, rice, and wheat, to eliminate the toxic effects on cultivated plants. This advancement will facilitate a deeper understanding of how these compounds influence weed physiology, enable the exploration of their metabolic pathways within weeds, and allow for the assessment of their efficacy and persistence. Such insights are crucial for ensuring ecological security and supporting the rational use of herbicides.”
Comment 2: Glyphosate was also used as a control when analysing herbicidal activity (Fig.8b,c). It is also necessary to show how plants grew on media with no substances added - a control is needed. There is a control in Figure 8, but what is it? Regular medium or medium with added glyphosphate?
Response: Thank you for your helpful comments. The test plants treated with glyphosate at 200 μg/mL were photographed, together with plants treated with concentrations of 100 and 80 μg/mL. In order to save space, they are not shown in Figure 8. The growth status of plants treated with glyphosate can be seen in Figures S18 and S19 in Supporting Information. The CK in Figure 8 is blank control, which is a regular medium without any substances added. We have made modifications and annotations in the revised manuscript (Lines 186-187).
Comment 3: Please provide figures on the inhibitory activity of the tested substances against bacteria.
Response: Thank you for your helpful comments. In Section 4.7, we have provided the method for antibacterial assay. The MICs were defined as the lowest concentrations of antibiotics with no visible growth of bacteria. Therefore, the MIC value is a numerical value and we didn't keep the photos/figures.
Comment 4: Does the endophytic fungus Aspergillus japonicus TE-739D have herbicidal and antibacterial properties? Why don't you use an endophytic fungus as an additional control?
Response: Thank you for your helpful comments. Yes, in our initial screening of endophytic fungi, the fungus Aspergillus japonicus TE-739D was found to possess promising antibacterial properties. Therefore, we chose this fungus to study its secondary metabolites. This study aimed to discover bioactive metabolites with lead compounds potential. Thus, we applied commonly-used drugs as positive controls, such as glyphosate for herbicidal assay and chloramphenicol for antibacterial assay, to compare their activities.
Comment 5: Minor Revisions see the pdf file.
Response: Thank you for your helpful comments. We have revised in the manuscript as suggested.
Reviewer 3 Report
Comments and Suggestions for Authors
This study represents and interesting study of two secondary metabolites isolated from the endophytic Aspergillus japonicus, associated with the plant Nicotiana tabacum. The team manages to isolate, identify and cultivate the A. japonicus strain and afterwards to isolate six metabolites numbered from 1 to 6. A big part of the study is dedicated to the structural elucidation only of the first two compounds and the other 4 were identified based on the literature, and weren’t objects of the structural elucidation. There is no reason pointed out about the reason for that. I have the following remarks and questions regarding this work:
Introduction
The introduction in informative enough but it could be enriched with more information regarding the biological activities of secondary metabolites and their potential applications.
M and M
The procedure applied for the isolation of the strain is unknown. There is no reference of previous study of the isolation of the strain used in the research.
Was the cultivation performed in the absence of light or there was specific light pattern involved?
For a non-specialist in the field, it is not clear what type of information is presented in subsection 3.5., and why the presented data is regarding only to compounds 1 and 2?
Why the authors perform seed germination inhibition assays only with compounds 1 and 2?
Results
Even though the unknown compounds were identified according to the literature why were they not further analyzed?
There is no discussion section!
Author Response
Comment 1: The introduction in informative enough but it could be enriched with more information regarding the biological activities of secondary metabolites and their potential applications.
Response: Thank you for your helpful comments. We have provided examples of the biological activities and potential applications of secondary metabolites in the introduction as suggested (Lines 32-45).
Comment 2: The procedure applied for the isolation of the strain is unknown. There is no reference of previous study of the isolation of the strain used in the research.
Response: Thank you for your helpful comments. We have added the procedure for the isolation and purification of Aspergillus japonicus TE-739D in Section 4.2 (Lines 283-293).
Comment 3: Was the cultivation performed in the absence of light or there was specific light pattern involved?
Response: Thank you for your helpful comments. The cultivation was performed in the absence of light as described previously (Reference 23, Microbiol. Res. 2021, 12, 829-839.).
Comment 4: For a non-specialist in the field, it is not clear what type of information is presented in subsection 3.5., and why the presented data is regarding only to compounds 1 and 2?
Response: Thank you for your helpful comments. The spectral data are crucial for the characterization of new compounds. Nuclear magnetic resonance (NMR) can resolve the planar structures and stereochemistry; Optical rotation data is used to identify the configuration of chiral compounds; Determining the absolute configuration of chiral molecules often use electronic circular dichroism; Mass spectrometry can clarify molecular weight and structural fragments; UV data helps to determine functional groups. By integrating these data, key information such as the structure and configuration of the new compound can be comprehensively and accurately determined. Moreover, in general, only the new compounds spectral data are needed to be characterized in the method section, while the known compounds are identified based on the literature data.
Comment 5: Why the authors perform seed germination inhibition assays only with compounds 1 and 2? Even though the unknown compounds were identified according to the literature why were they not further analyzed?
Response: Thank you for your helpful comments. The new compounds possess a novel chemical structure, and studying their biological activities is expected to discover new therapeutic targets and mechanisms of action. For the known compounds, their activities have been partially clarified, and when new requirements arise (such as discovering new uses) or new theories emerge, further research of the known compounds will be conducted. In addition, the known compounds are identified according to the literature in the field of natural products chemistry and no further analysis is required (see References 1, 2, 5, 6, 7, etc.).
Comment 6: There is no discussion section!
Response: Thank you for your helpful comments. We have added the discussion section in the revised manuscript (Lines 205-262).
Round 2
Reviewer 1 Report
Comments and Suggestions for Authors
The points raised by this reviewer has been adequately dealt with.
Reviewer 2 Report
Comments and Suggestions for Authors
The authors have taken into account all my suggestions and made appropriate changes to the manuscript. I believe that the manuscript is acceptable.
Reviewer 3 Report
Comments and Suggestions for Authors
The authors concidered the marks on the manuscript which is now acceptable.